# Perception of Women’s Knowledge of and Attitudes towards Cervical Cancer and Papanicolaou Smear Screenings: A Qualitative Study in South Africa

**DOI:** 10.3390/healthcare11142089

**Published:** 2023-07-21

**Authors:** Zintle Gwavu, Daphne Murray, Uchenna Benedine Okafor

**Affiliations:** 1Department of Public Health, University of Fort Hare, 5 Oxford Street, East London 5201, South Africa; zintleg7@gmail.com; 2Department of Nursing Science, University of Fort Hare, 50 Church Street, East London 5201, South Africa; dmurray@ufh.ac.za; 3Faculty of Health Sciences, University of Fort Hare, 5 Oxford Street, East London 5201, South Africa

**Keywords:** knowledge, attitude, cervical cancer, Pap smear screenings

## Abstract

Background: Cervical cancer is the most common form of cancer worldwide. Consequently, it is crucial that women are encouraged to undergo interventions early via Papanicolaou (Pap) smear screenings to improve their health. In light of this, this study explored the knowledge of and attitudes towards cervical cancer and Papanicolaou (Pap) smear screenings among women in the Caleb Motshabi district, South Africa. Four focus group discussions were carried out among 19 women. The interviews/discussion were audio-recorded and transcribed verbatim and then analysed thematically. In this regard, seven main themes emerged that provided insight into the perceptions of the participants regarding their knowledge of and attitudes towards cervical cancer and Papanicolaou (Pap) smear screenings. While the majority of participants were aware of cervical cancer and Pap smears, they lacked more specific knowledge of what this cancer is or its related causes. Although some participants had had a Pap smear done, they neither knew how the procedure was done nor the reasons for it. In addition, most mentioned receiving information about this procedure from their peers rather than healthcare workers. Notably, those with previous experience indicated that they had regular screenings. Furthermore, they better understood exactly how it is done. The findings emphasised women’s limited knowledge of cervical cancer and Pap smears. It further highlighted the need for sustainable education programmes and mobile clinics to encourage an awareness of and accessibility to this particular type of screening within South African communities. Therefore, intervention strategies that make people aware of this specific cancer and encourage the uptake of Papanicolaou (Pap) smear screenings are crucial, as is the continued advocacy for sustained educational programmes and accessible healthcare services.

## 1. Introduction

According to worldwide statistics, cervical cancer is the fourth most commonly diagnosed cancer and the fourth primary cause of cancer-related mortality in women, with an approximated 604,000 newly identified cases and 342,000 deaths in 2020 [1]. Also, in contrast with developed nations, females in less developed countries have substantially higher cervical cancer mortality rates (12.4 per 100,000 versus 5.2 per 100,000) [1]. On a regional basis, cervical cancer is a major cause of cancer-related mortality in 36 nations, and nearly all of these countries are located in sub-Saharan Africa, including the Republic of South Africa [1].

Research has established that various factors are associated with the development of cervical cancer, including high-risk human papillomavirus infections (HPV types 16 and 18), being sexually active from a young age, a higher number of pregnancies/childbirths (high parity), having multiple sexual partners, human immunodeficiency virus (HIV) co-infections, immunosuppression, and specific dietary deficiencies [2]. Other significant precursors include certain sexually transmitted infections (HIV and Chlamydia trachomatis), smoking, and long-term oral contraceptive usage [3]. Previous studies linked one of the most common causes of HPV and 90% of invasive cervical cancer cases to precancerous changes in the cervix; however, owing to the extremely effective primary (HPV vaccine) and secondary (screening) preventative measures against HPV, cervical cancer is thought to be essentially preventable [1]. Similarly, the worldwide incidence of new cancers of the cervix attributed to HIV among women living with HIV is 6% [4,5]. South Africa’s high prevalence of HIV, which is a risk factor for cervical cancer, is concerning since current statistics indicate that it is the third most common cancer affecting women of reproductive age [6]. In 2018, it accounted for more than 560,000 new diagnoses per year, with over 311,365 deaths. These numbers translate to approximately one death every two minutes per year on a global scale [7]. The country’s socioeconomic status is another factor that contributes to its high prevalence of HIV/AIDS.

In South Africa alone, it is estimated that a staggering 7.52 million people live with HIV. Moreover, 4.7 million are women [8]. Previous research indicated that HIV-positive women are three to five times more likely to develop cervical lesions that can become cancerous than those who are HIV-negative. Furthermore, 51% of South African women are living with HIV/AIDS [9], which is a statistic implying that it is a high-risk environment for the acquisition of cervical cancer. Despite many of these women receiving antiretroviral drugs, there is conflicting evidence around whether these medications effectively reduce their risk of cervical cancer. Consequently, it is of utmost importance to prevent this disease, nationally and globally [10].

However, earlier research seems to indicate that the South African Department of Health’s targeted 70% cervical cancer screening projection is unachievable since only about 13.6% of this is currently being attained resulting in a high cervical cancer mortality rate [11].

Despite awareness campaigns highlighting the disease, there is a poor uptake of cervical cancer screening in African countries. Seemingly, a general lack of knowledge, ignorance, and misinformation about how the procedure is performed are the likely mediating factors for the low cervical cancer screening in resourced-constrained settings. Therefore, the present study explored South African women’s knowledge of and attitudes towards cervical cancer and Papanicolaou (Pap) smear screenings.

## 2. Materials and Methods

### 2.1. The Setting, Study Design, and Participants

This study was conducted in the Caleb Motshabi district of Bloemfontein, which is the capital city of the Free State province in South Africa. This particular province has five districts, namely, Thabo Mofutsanyane, Fezile Dabi, Motheo, Xhariep, and Lejweleputsua. Caleb Motshabi, in particular, is a township south of Bloemfontein and is home to approximately 7500 households. A community church was utilised as a venue in which the interviews/discussions took place. Concerning the methodology, this study utilised a qualitative, explorative design, employing semi-structured focus group interviews with nineteen (19) purposively selected Black South African women aged between 18 and 60. Based on the South African Cervical Cancer Policy, this population was appropriate since it represented those eligible for free screening who were legally of age. Moreover, this study focused on this particular demographic due to the high incidence of cervical cancer associated with it. This age group of women presents the highest disease incidence in the country.

### 2.2. Data Collection Procedure

This study utilised a semi-structured interview guide containing open-ended questions informed by the research objective to collect data and conducted four focus group interviews, consisting of four to six participants. The data were collected until data saturation was reached. In addition, interviews were conducted in either English or a local language. The latter was later translated into English.

While the abovementioned interview guide was pretested in a pilot study of five participants to ensure its content validity, its findings were not included in the actual study. Once participants had given the researcher permission to audio-record their interviews, the discussions were held. Each lasted about 40 to 60 min, after which, they were transcribed and compared with the researchers’ written notes.

A total of four questions were asked, namely:What do you know about cervical cancer? (From your understanding, what would you say it is?)What do you know about the Pap smear? (How is the procedure done?)What is the purpose of a pap smear?What advice would you give to the Department of Health regarding cervical cancer and Pap smears?

### 2.3. Trustworthiness

The study maintained its trustworthiness by applying the concepts of credibility, dependability, auditability, conformability, and transferability throughout its duration [12]. Furthermore, bracketing ensured that all previously known knowledge and beliefs about cervical cancer and Papanicolou smears were set aside. The researcher further made use of member checking and an audit trail established the data’s validity and reliability. Finally, a professional independent co-coder (external) assisted with validating the data, as applicable to qualitative research.

### 2.4. Data Analysis

The audio-recorded interviews and field notes were transcribed verbatim, after which, data were analysed by applying the steps described by Tesch [13]. To obtain an overall sense of the discussions, the researcher read through the transcripts. Next, they compiled a list of all topics and similar topics by employing clustering. After this, they listed all relevant points and grouped them into various thematic categories. Finally, the researcher, authors, and coder agreed on how to analyse and present the study’s findings.

### 2.5. Ethics

The Human Research Ethics Committee of the University of Fort Hare (Ref # 2021=03=04=GWAVU) granted ethical approval for this study. In addition, the Eastern Cape Provincial Department of Health Ethics Research Committee approved its protocol and permitted researchers to conduct it in selected health facilities. The ward councillor and the pastor of the venue also provided their permission for the venue to be used for interviews. The participants were then informed of their voluntary participation and that they could withdraw at any time without prejudice. Furthermore, they were assured of their anonymity and their right to privacy and confidentiality.

## 3. Results

### 3.1. Demographic Profile of the Participants

The participants’ ages were between 18 and 60 years. Of the 19 participants, four attained a high school diploma, while none had a postsecondary degree. The majority of the participants were single, and nearly half were employed on short-term contracts.

The major themes and sub-themes that emerged from the participants’ interviews regarding their knowledge and attitudes on cervical cancer and Papanicolau smear screenings are summarised in Table 1 and Table 2 below.

### 3.2. Theme 1: Knowledge of Cancer

Most participants seemed to have heard of the term “cervical cancer”, but were unclear as to what it was. As a result, the participants answered based on their own understanding of “what it is”. The researcher established that the majority of those who had previously heard of it had either received the information from friends or during clinic visits.

#### 3.2.1. Sub-Theme 1.1: Participants’ Conceptualisation of Cancer

As mentioned above, participants’ own understanding of the disease was based on information they had received from others, and none had been formally educated on the topic. In general, they knew it is a disease of a woman’s private parts or womb. In addition, most had heard about it from friends.

These are some of their responses:

“Cervical cancer, I think, it is the cancer of the womb; that’s what I know.” (Focus group discussion (FGD 3).

“I’m thinking that maybe there’s a sore in the womb that is not supposed to be there, and it needs to be treated, so that it is not there. I think there’s something like a sore in the womb.” (FGD 1).

“It’s the cancer in women. It’s found in women on their private parts.” (FGD 4).

#### 3.2.2. Sub-Theme 1.2: Contributory Factors for Cancer

Regarding factors contributing to cervical cancer, the respondents identified sexual activity as one of the most common factors related to this particular cancer. In addition to this, a few further mentioned multiple partners, using different types of soaps with scents, failing to change pads during menstrual period days, diet, and smoking.

The following comments reflect the above:

“I think its sexual activity.” (FGD 2).

“I feel like the soaps we use to wash as some are scented, and other are not, and the pads we use some are scented, and I think they are the ones that leave that dirt and the soaps we use it’s them.” (FGD 1).

“A woman in their tubes like because they are hot, they must regularly wash, and when they put on pads, don’t use one pad the whole day; you must change it; so it’s caused by such, when you don’t change it, you must wash.” (FGD 2).

“I think it’s the food we eat; it’s not like the food from long ago. I think they are contributing.” (FGD 3).

“Smoking and stress.” (FGD 1).

Although participants had some clue that sexual activity can cause cervical cancer, they were unaware of the manner in which it did so. None mentioned HPV as a cause of cervical cancer, indicating that they did not know about it. Since most participants had children, it could be assumed they were, or are still, sexually active. Consequently, their lack of knowledge of HPV is of great concern.

### 3.3. Theme 2: Knowledge of Pap Smears

Although most participants had knowledge of cervical cancer, albeit limited, they knew about Pap smears. While some had only heard about it, others had undergone the procedure and knew how it is performed. The information that they received from their friend impacted whether they considered undergoing the procedure.

#### Sub-Theme 2.1: Participants’ Perceptions of Pap smears

Only six of the nineteen participants had undergone the procedure. Of these, three underwent a routine procedure at a clinic during an antenatal visit, while three had done so after an awareness programme. Of the six, two participants never received their results, while four had. The thirteen participants who had never had a Pap smear were not interested in having one or were too scared to go. However, this changed after the discussions.

In regard to this sub-theme, the following responses were noted:

“It’s not something painful; it’s they just insert something in your womb and take sample and test it, so they can see if you have cancer; that’s how I knew about it.” (FGD 3).

“Not interested; it’s because others who have done Pap smear say the wood used to do it, can be painful. I think it’s that fear that, as a woman, you know the pain of giving birth, so you get scared that what this person has told me is the way I’ll also feel it, and you get scared, ma’am.” (FGD 1).

“No, I’m scared. Someone, a friend, said they insert a metal, when I asked how it’s done. When she explained that, I just decided I’ll see it later. I never asked after inserting the metal what happens; I just felt my blood running.” (FGD 2).

“I haven’t gone to do the test because I’m scared since my mother has cervical cancer.” (FGD 3).

The lack of information received by qualified healthcare professionals had created a negative perception of how the procedure is done. As a result, there was low uptake of Pap smear screenings. Nonetheless, creating an awareness of these procedures may help to eliminate this issue and empower women to better care of their own health.

### 3.4. Theme 3: Knowledge Deficit

Participants’ levels of knowledge of cervical cancer and Pap smear screenings were low. Out of a total of nineteen participants, only six had undergone the procedure. Their lack of knowledge was also demonstrated by the fact that none mentioned HPV as a risk factor, even though it is the primary cause of cervical cancer.

Participants mentioned being concerned about the lack of information provided by clinic nurses, the absence of Pap smear screening campaigns, and receiving unclear information from peers. While the Batho Pele principles state that people have the right to information, participants were adamant that this was not being fulfilled. In addition, they said they were scared to ask the nurses any questions because of the negative response they would receive.

The following comments support this:

“I want to do it because I heard my boyfriend’s aunt died because of it, and it had reached the breasts. That’s why when I heard about this meeting, I decided to come and hear for myself and not from people, but from people who have knowledge.” (FGD 1).

“Sometimes you know, sister, when they talk to us speak, speak like we, not human beings, because they shout; that’s why some other people do not ask because when you ask, the person shouts, and they have the information. They can’t be calm and tell you ‘step number 1: you do this and that’, that’s why people are scared to do a Pap smear.” (FGD 3).

“99% of people going to clinics are not treated nicely, and they end up not taking their medications. That’s why you lucky if you go to a place simply going to give birth, and you don’t get good treatment; you lucky if you go to a government clinic and you get good treatment; that’s why many people are scared to go do Pap smears it because of the attitude they will get from those people.” (FGD 2).

The participants made it clear that when they attend clinics, the attending nurses tell them to undergo Pap smear screenings but fail to explain their importance. Women should receive health education regarding these topics from the nurses in their local clinics, with complete information at their disposal to make informed decisions about their health.

### 3.5. Theme 4: Attitude to Pap Smears

Attitude is essential to how a person behaves; as such, it impacts many health determinants and makes it difficult to change someone’s health-related behaviours [14]. However, participants stated their willingness to undergo such screenings after the discussions. Their initial negative attitudes changed when they heard from those who had already been screened. However, two participants were still sceptical about having the test done.

#### 3.5.1. Sub-Theme 4.1: Positive Attitude Regarding Doing Pap Smears

The purpose and importance of Pap smear screenings resulted in many participants being interested in undergoing one themselves. In addition, those who had not been screened were keen to do so after hearing from participants who had. They further indicated that they wanted to be tested for cervical cancer because they wanted to know where they stood with regard to their health, particularly in terms of cervical cancer. Furthermore, the six participants who had been screened were willing to be tested again because of the associated benefits.

Participants mentioned the following:

“Because I want to know where I stand and what’s going on. Do I have a problem cause now I don’t know if there’s a problem or not?” (FGD 2).

“No, I haven’t done it, but I would be happy to do it. I even thought that today I would be doing it. I even washed thinking we were going to do it.” (FGD 1).

“Yes, I would because, as we were discussing, the womb can have many diseases, and they can even find if there’s infections.” (FGD 3).

This means that if the women are well educated about cervical cancer and Pap smear screening, their attitude will affect the uptake of the screening test.

#### 3.5.2. Sub-Theme 4.2: Negative Attitudes Regarding Doing Pap Smear Screenings

Lack of information coupled with misinformation seemed to fuel the participants’ negative attitudes towards the screening test. If an individual does not know the importance of the screening test and what health benefits it has on their overall health status, they can be influenced negatively. This is why having awareness programmes regarding cervical cancer and Pap smear screenings would inform the community and eliminate any misinformation shared among their peers.

Responses from participants included the following:

“Since I booked, I never went. I think I’m lazy.” (FGD 2).

“No, I did not because I was scared. I felt like it’s painful though they said it’s not painful, and they said they won’t force anyone to do it.” (FGD 3).

“No, I won’t do it. I don’t like doctors and clinics.” (FGD 3).

While a lack of information makes an individual inactive in areas related to their well-being, when they are fully equipped with the knowledge of both the pros and cons of specific tests/procedures, it is easier for them to be proactive.

### 3.6. Theme 5: Challenges of Undergoing Pap Smear Screenings

According to [15], non-compliance with follow-up appointments after a Pap smear is another reason for its poor prevention. This was evident from some of the comments regarding follow-ups of Pap smear tests. Failing to follow up means that even if the test found an abnormality that is treatable, it is not attended to. Further challenges include factors such as delayed results and Pap smears not being prioritised when healthcare workers conduct community awareness programmes.

#### Sub-Theme 5.1: Hostile Attitude of Nurses

Participants displayed a lack of confidence in their public health facility nurses and thought the nurses did not care much because of their bad attitudes. In addition, these nurses seemed to neglect patients since they were rude and chatted constantly while people were waiting to be assisted. Furthermore, having to spend more than five hours at the facility, not being attended to, or the absence of being given information is problematic. Participants indicated that if they could have afforded private healthcare, they would have rather made use of that option.

The following comments were made:

“I would agree with no5 because things from the government are not like the private because if I ‘didn’t go to private, I don’t think I would’ve received my results; it’s like they don’t care.” (FGD 4).

“No, they don’t explain before you agree to do it; only when you are inside, will they explain” (FGD 2).

“…cause in the clinic you can spend 5 h without being attended to.” (FGD 3).

### 3.7. Theme 6: Recommendations

At the end of the focused group discussion, participants were asked to make recommendations to the Department of Health in terms of how it should proceed with the province’s cervical cancer and Pap smear screenings. The recommendations that were made are summarised below:
To implement health education talks at clinics;Prioritise cervical cancer;Intensify awareness programmes;Do early and regular testing for diagnoses;Have mobile clinics that assist with testing;Identify a particular day in healthcare facilities focusing on Pap smear tests;Doing door-to-door campaigns;Providing patients with transport to healthcare centres;Lastly, give sound, clear explanations about the procedure when it is performed.

Should these recommendations, where possible, be implemented, the Department of Health would be able to overcome the low screening uptake in the province. In this regard, the details of these recommendations are presented as sub-themes with comments below:

#### 3.7.1. Sub-Theme 6.1: Health Education Talks at Clinics

Most participants who had heard about Pap smears or undergone the test stated that it was done on an opportunistic basis during their clinic visits, whether while attending antenatal care or having their children immunised. Nevertheless, there is a distinct lack of effort to educate patients about the test and its importance. Several participants recommended that nurses purposely use such opportunities to present educational talks to increase the uptake of these screening programmes.

Participants in the first focus group made the following two comments:

“At the clinic, they must talk to us and tell us how to prevent cervical cancer.” (FGD 1).

“Even in the family planning clinics, the nurses don’t explain these things to us because we go for family planning, but we just get the tablets OR injections, then we leave. They must educate us!” (FGD 1).

#### 3.7.2. Sub-Theme 6.2: Cervical Cancer Must Be Prioritised

One of the issues raised by the participants is how the topic of cervical cancer screening is neglected during awareness programmes. Participants suggested that when healthcare workers are screening tests for other medical conditions like hypertension and diabetes, Pap smears should form part of the screening to increase uptake.

Comments by participants:

“In terms of cervical cancer and Pap smears, they would prioritise it just like other sicknesses because of the diseases they always discuss.” (FGD 4).

“Make it a priority like other sicknesses.” (FGD 3).

#### 3.7.3. Sub-Theme 6.3: Awareness Programmes to Be Intensified

Awareness campaigns should make use of media, such as local radio broadcasts or pamphlets and/or posters, using local, community languages and should be clearly displayed in local areas. According to Touch and Oh [16], using many diverse media channels could greatly impact and increase communities’ awareness of cervical cancer.

One participant made the following comment:

“They must have awareness programmes in the community so that we, as the community, could be educated regarding cervical cancer and Pap smear screening.” (FGD 1).

According to [17], introducing focused and comprehensive, community-based awareness programmes on Pap smears, along with physicians’ recommendations for patients to undergo Pap smear screenings, would greatly increase the uptake of screenings. In contrast, communities that lack knowledge of the above will present with low numbers of screenings until they introduce appropriate interventions.

#### 3.7.4. Sub-Theme 6.4: Do Early and Regular Tests for Early Diagnoses

Participants identified that being educated about cervical cancer would equip them to get screened so they can know their health status early. In addition, by educating the community when the DoH offers free screening tests and under which conditions a person can have the tests done regularly, people would know at which age and how many times they are entitled to the free tests.

Two relevant comments by participants were as follows:

“I think regular tests every year can prevent it, but if we sit and not go, they ‘can’t treat it like my mom; they found it late, and they said if it had been caught earlier, she could’ve been treated.” (FGD 4).

“To limit the spread of cervical cancer because if a lot of people can do it, they’ll know and be treated early, so the more the people can do it can be limited.” (FGD 3).

If women can be motivated to undergo screenings through awareness programmes and if healthcare professionals provided them with adequate information about their importance, the incidences of cervical cancer in South Africa would decrease.

#### 3.7.5. Sub-Theme 6.5: Mobile Clinics Should Be Equipped to Perform In Situ Pap Smears

Participants suggested that mobile clinics be sent out to conduct Pap smear tests within communities since clinics were sometimes situated very far from where the women resided. In this regard, several mentioned that they attended a clinic approximately 6 km away from their home and that traveling to the clinic was sometimes an issue. To address this issue, a few participants suggested that mobile clinics should conduct community-based Pap smear tests, possibly on a monthly basis, to close this gap and increase the uptake of such screenings.

Participants of the second FGD commented as follows:

“Bring services to us so they can test Pap smears and sugar and other stuff, so we can know where we stand.” (FGD 2).

”For them to make mobile clinics that offer services like Pap smears and other things because when you go to the clinic to speak about your issues of this and that, referring to something from Wednesday, they say: no, talk about what the issue you have today.” (FGD 2).

The above comments indicate that sending mobile clinics sent to communities would help to increase the number of Pap smear screenings since that has been the case with other screenings presently offered in this manner, such as HIV and pregnancy tests.

#### 3.7.6. Sub-Theme 6.6: Identifying Days That Focus Specifically on Pap Smear Screenings

If clinics dedicate specific days known to the community as reproductive health days, participants would feel motivated to undergo related screenings. However, they report that when attending a clinic, they are only told to write their names down if they want a Pap smear done, but were not informed whether there were weekdays for such tests.

Comments by participants included the following:

“It’s the whole chaos of the clinics.” (FGD 2).

“It’s like a problem with the clinics; it’s that line will be there and the process of doing a file just to do a Pap smear. And before it was made easy, you were just going to the door where they are doing Pap smears at Pelonomi hospital, and you would do it, you ‘wouldn’t spend a lot of time there.” (FGD 2).

From the above, it is clear that the women are of the opinion that having certain days to attend to gynaecological matters would be invaluable to communities, where having specialised procedures, such as baby immunisation and antenatal care, that were day-specific would make it easier for patients to attend.

#### 3.7.7. Sub-Theme 6.7: Door-to-Door Campaigns Necessary

As part of creating awareness, door-to-door campaigns were suggested. For example, the DoH could educate community healthcare workers on Pap smear screenings, which would further educate their related communities. This may prove important since there are not enough healthcare workers in the country. In addition, door-to-door campaigns and educating people in their homes would likely increase the uptake of said screenings.

Comments by participants:

“Make door-to-door campaigns like they do with TB and HIV.” (FGD 3).

“If they can do door-to-door because now we hear through other people without having full information because they check many diseases they can check.” (FGD 3).

#### 3.7.8. Sub-Theme 6.8: Provide Transport to Healthcare Centres

As mentioned previously, participants stated that their local clinic was at least six kilometres from their residences and that most community members were unemployed. Another problem they highlighted was one of transport. In this regard, various women suggested having a patient transport system to drive patients from the community to the clinics and back if the Department of Health wanted to see an increase in Pap smear screenings.

Since the majority of participants were unemployed, money, or the lack thereof, was seen as a barrier to them attending health screenings. If patient transport could be introduced, it would motivate the community to attend such screening programmes and related follow-ups.

Two comments, in particular, highlighted this sentiment:

“We don’t have a problem doing it, but hospitals and clinics delay, and sometimes you don’t transport you going to the clinic you travel a long distance, and you don’t have money.” (FGD 2).

“Clinic is from here till around Shoprite, (about 5/7 km), that’s the problem.” (FGD 2).

It is clear that these women feel that the clinic’s distance and their own lack of money played a big role in whether they utilised the provided healthcare services. They further stated that if the DoH wanted to motivate people to attend the screening services, patients should be met halfway since most South African citizens are unemployed. Therefore, it is vital to explore various measures to increase these screening services’ uptake.

#### 3.7.9. Sub-Theme 6.9: A Clear Explanation of the Testing Procedure Should Be Given

Participants suggested that healthcare workers educate them on the procedure while performing it since this would eliminate patients’ fear and calm them down. If an individual has a good experience, they are likely to mention it to their community counterparts. This would then lead to more people voluntarily undergoing such tests. In this regard, women’s understanding of the screening process influences the uptake of Pap smear testing [14]. Consequently, it is important that an individual is educated on the process because they are likely to spread this knowledge to others in their communities, potentially sparking their interest.

Relevant comments by participants include the following:

“I think they can do Batho Pele principles; everyone has the right to ask those who are going to get tested; they must be able to ask whatever and be given step by step. Some people are scared to ask, like when you will get your results.” (FGD 4).

“They should have room in the clinic where someone will explain and teach you about it too before you decide to do it.” (FGD 4).

Most participants in this study reported a lack of information from nurses regarding Pap smear procedures.

## 4. Discussion

### 4.1. Knowledge About Cancer

According to the findings above, there was a lack of knowledge regarding cervical cancer among the interviewed women. This was demonstrated by the participants being unable to define cervical cancer. Instead, when prompted, they gave their opinions of what it is. All participants in the study presented such ignorance and it was noted that none could define cervical cancer adequately. While some knew it was a disease related to the womb or women’s private parts, others could only describe the disease as “pimples” in the womb or cancer of the private parts. When asked about the related risk factors, the most popular response was sexual activity and multiple sexual partners. However, they did not know how sexual activity or multiple sexual partners causes cervical cancer. This supports the findings of a similar study in Gabon, which revealed participants’ poor knowledge due to an ongoing lack of educational interventions about cervical cancer [18]. A recent study found an increased risk of HPV-related tumours for cancer in extra-cervical sites following treatment for intraepithelial neoplasia of the cervical (CIN 2–3), arguing for the implementation of individualised follow-up and screening initiatives [19]. Educational intervention is an essential approach to addressing the above issues. As mentioned, the DoH must address this common problem if it wishes to see a reduction in the disease’s related mortality rate due to cervical cancer. Another Kenyan-based study explored educational interventions among women in terms of this particular disease. Some topics included in the discussion were how a Pap smear is done, the outcome of the screening, health facts about cervical cancer, related risk factors, and treatment options. The discussion was led by community health workers who had attended training on how to teach to better equip them for the task they had to undertake [20].

### 4.2. Participants’ Knowledge of Pap Smears

Most participants had some idea of what a Pap smear is, even though their knowledge was not detailed. Several described it as an instrument inserted in the womb or done to check if the womb is normal. Most had heard about the procedure from other people, while their second most common source of information was nurses. The information received from the former mentioned that something was inserted in the woman’s private area. However, participants had differing views of the procedure. For example, clinic nurses just informed the participants of a Pap smear, but no further information was given. This contradicts a similar qualitative study about cervical cancer screening in South Africa that found that most participants did not get information about Pap smear screening from peers in their communities, and the few who had heard from peers had heard negative information that made them not have the screening.

It further noted negative comments, such as doing a Pap smear would make participants worry [21]. This does not support the findings of the current study, in which most participants had reported hearing about the procedure from peers. Both studies similarly reported the negative information participants received discouraged them from attending screenings because of fear. However, Pap smear screenings were reported to save lives by reducing invasive cervical cancer incidences and decreasing the advanced stage of the disease. In fact, countries with successful screening programmes above 70% have a death rate of less than 2 per 100,000 women per year [22], and other countries, like the United States, have seen a decrease in cervical cancer rates of 70% over the last 40 years. This indicates that when women attend screening programmes, they increase their chances of having cervical cancer detected early, and thus, are more likely to present favourable outcomes in terms of treating the abnormal cells.

### 4.3. Knowledge Deficit

Clearly, as seen by the participants’ comments, this particular community lacked knowledge of both cervical cancer and Pap smear screenings. Furthermore, it should be noted that being aware of a certain health issue does not necessarily mean that an individual will make informed decisions about their health. Although sufficient information is of importance, the main challenge participants raised was the lack of campaigns on such matters. It seems that although nurses have the platform to present health talks during patients’ visits, they are not doing so, and as a result, participants receive unclear information from their peers.

Since well-educated women tend to be highly knowledgeable of both cervical cancer and Pap smear screenings, it is believed that education is an essential predictor of the levels of awareness [23]. However, most participants in this study had not completed their high school qualifications, which could be why they lacked an adequate level of knowledge pertaining to the cancer screening programmes. Consequently, educating women would lead to increased uptake of the screening test; therefore, patients should be educated on vulvar self-examination for reporting minor and major vulvar changes, and cervical cancer screening visits should be viewed as a crucial opportunity for cancer prevention [24]. Nevertheless, this will not happen overnight, as seen in a study in Nigeria in which educational interventions to increase knowledge about Pap smear screenings did not yield positive results after the intervention [25]. Despite this, a constant supply of information may help to convince women to develop an interest in screening to be aware of their health status. In order to reduce the impact of HPV-related malignancies, educational intervention should be centered on vaccination and screening efforts; recent multi-society statements have reinforced this concept [26,27].

### 4.4. Attitudes towards Pap Smears

The participants in this study who had undergone the screening test generally had a positive attitude, in contrast with those who had not, who displayed a negative attitude. Understandably, those with previous experience had first-hand experience of how it is performed, and therefore, are less fearful of it. In contrast, those without experience received their information from hearsay, which may contribute to their fear. Therefore, receiving a first-hand, positive review of a particular procedure may influence an individual’s view of it, while the same can be said about a negative review.

In this study, participants’ positive view of Pap smears was attributed to the clinic handling the procedure well, while general interest in having the procedure was linked to previous positive experiences. Negative attitudes were linked to a lack of interest in it, the procedure being uncomfortable, and/or a fear of the unknown. In Botswana, a similar study indicated that 95% of its participants would undergo a Pap smear if given the chance. The reason was that the participating women were interested in their cervical health and wanted to improve their health overall. They further indicated that if diagnosed with cancer, they would initiate early treatment [28]. In addition, participants of a Kenyan study were also positive about undergoing a Pap smear because the healthcare workers doing the examinations were polite to them [29]. This highlights how healthcare workers’ attitudes influence women’s motivation to be tested.

### 4.5. Challenges Experienced during Pap Smears

Participants reported negative experiences when visiting clinics for cancer or other medical appointments; these negative experiences could be viewed as potential barriers to their adoption of cervical cancer screening and vaccination. The combined vaccination and screening strategy was shown to be cost-effective in several low–middle-income countries [30,31,32], suggesting that it may be possible to prevent cervical cancer in contexts with limited resources. As previously stated, the majority of the participants in this present study cited challenges relating to education and awareness (lack of schooling, community education efforts, and awareness campaigns), health services organisation (inadequacies related to poor clinic organisation, lack of dedication to women’s health days, and lack of mobile clinics and transportation), and poor quality of health services provision (negative and poor attitudes of healthcare providers). However, the majority of problems were due to human error, and therefore, can be addressed by adopting appropriate measures in the clinical setting. Therefore, a collaborative and multi-sector strategy is needed to mitigate these challenges. Government, schools, and non-governmental organisations concerned with women’s health should create health education advocacy and awareness in schools and communities, as well as organise health talks and road shows on the significance of cervical cancer screening. In this context, the Department of Health ought to strengthen the existing community-wide awareness campaigns for additional illnesses and integrate cervical cancer awareness into its awareness program. Additionally, the educational programs, whether administered in clinics or as outreach programs, will raise community awareness about cervical cancer and emphasise the risk factors associated with the disease. Furthermore, the availability of sufficient resources, as well as trained and committed personnel, would aid in promoting cancer screening. Finally, considering that this was a qualitative study that sought to provide insights, further studies that include cross-sectional quantitative studies or population surveys on some of the interventions proposed here to address the context-specific challenges of cervical cancer screening in this low-resource geographical setting would provide more comprehensive and accurate perspectives on the effectiveness or implementation of these interventions.

## 5. Limitations of the Study

The limitations of this study are worth noting. Caleb Motshabi was the only location in Bloemfontein where the study was administered; consequently, all participants were from this location. Due to the small sample size and the fact that only one race participated in this study, it is not possible to assume that all races in this context share the same perspectives. Furthermore, these results cannot be extrapolated to the entire Free State Province or the entire country. In addition, considering that the present research was a qualitative investigation that aimed to provide insights, additional quantitative cross-sectional studies or population surveys, as well as context-specific strategies about cervical cancer screening, are required. Despite the dearth of available data, this study provides some insight into the perspectives of cervical cancer screening in a region with scarce resources.

## 6. Conclusions

The women lacked knowledge and had undesirable attitudes regarding cervical cancer and Pap smear screening. Therefore, context-specific interventions to improve the uptake of cervical and Pap smear screenings are advocated for.

## Figures and Tables

**Table 1 healthcare-11-02089-t001:** Demographic profile of the participants (n = 19).

Variables	Number of Participants
**Age (years)**	
20–30	3
31–45	11
45–60	5
**Level of education**	
Grade 7–9	8
Grade 10–11	7
Grade 12	4
**Marital status**	
Married	3
Single	15
Widowed	1
**Employment status**	
Employed	7
Unemployed	12

**Table 2 healthcare-11-02089-t002:** Identified themes and sub-themes.

Themes	Sub-Themes
Knowledge of cancer	• Participants’ conceptualisation of cancer. • Contributory factors for cancer.
Knowledge of Pap smears	• Participants’ perceptions about Pap smears.
Knowledge deficit	• Lack of knowledge of participants.
Attitude to Pap smears	• Positive attitude regarding doing Pap smears. • Negative attitude regarding doing Pap smears.
Challenges experienced while undergoing Pap smears.	• The hostile attitude of nurses.
Suggestions/recommendations	• Health education talks at clinics. • Cervical cancer must be prioritised. • Awareness programmes about it are to be intensified. • Early and regular testing for early diagnoses. • Mobile clinics be sent out to do Pap smears. • Identifying days, especially focusing on Pap smears. • Door-to-door campaigns are necessary. • Provide transport to healthcare centres. • Clear explanations should be given when performing Pap smears.

## Data Availability

Data available on request.

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
