# Peer review of "Perception of Women’s Knowledge of and Attitudes towards Cervical Cancer and Papanicolaou Smear Screenings: A Qualitative Study in South Africa"

_healthcare, 2023, doi:10.3390/healthcare11142089_

Round 1

Reviewer 1 Report

Dear Authors,

this is an interesting paper on a crucial issue for women's health and public health policy. However some points can be improved.

- page 10, paragraph 4.1: about knowledge about cancer, please expand discussion also on risk of second primary cancer in cervical cancer screening context (please cite Preti M, Rosso S, Micheletti L, Libero C, Sobrato I, Giordano L, et al. Risk of HPV-related extra-cervical cancers in women treated for cervical intraepithelial neoplasia. BMC Cancer. 2020;20(1)., which highlights the risk of other cancers caused by HPV). This should be reinforced in order to further advance knowledge in the public.

- page 11, paragraph 4.3: the cervical cancer screening visit should be an important prevention moment: also patients should be educated about vulvar self-examination for reporting minor and major vulvar changes (please cite: Preti M, Selk A, Stockdale C, Bevilacqua F, Vieira-Baptista P, Borella F, et al. Knowledge of Vulvar Anatomy and Self-examination in a Sample of Italian Women. J Low Genit Tract Dis. 2021;25(2):166–71. )

page 11, paragraph 4.3: educational intervention should be oriented in order to diminish the impact of HPV related cancers (recent multi society statements reinforced this concept: 

Preti MJoura EVieira-Baptista P, et al The European Society of Gynaecological Oncology (ESGO), the International Society for the Study of Vulvovaginal Disease (ISSVD), the European College for the Study of Vulval Disease (ECSVD) and the European Federation for Colposcopy (EFC) consensus statements on pre-invasive vulvar lesions Kesic VCarcopino XPreti M, et al The European Society of Gynaecological Oncology (ESGO), the International Society for the Study of Vulvovaginal Disease (ISSVD), the European College for the Study of Vulval Disease (ECSVD), and the European Federation for Colposcopy (EFC) consensus statement on the management of vaginal intraepithelial neoplasia    

English is overall good.

Author Response

Comment:

- page 10, paragraph 4.1: about knowledge about cancer, please expand discussion also on risk of second primary cancer in cervical cancer screening context (please cite Preti M, Rosso S, Micheletti L, Libero C, Sobrato I, Giordano L, et al. Risk of HPV-related extra-cervical cancers in women treated for cervical intraepithelial neoplasia. BMC Cancer. 2020;20(1)., which highlights the risk of other cancers caused by HPV). This should be reinforced in order to further advance knowledge in the public.

Response:

The above-mentioned citation has been included in the discussion as suggested thus: A recent study found an increased risk of HPV-related tumours for cancer in extra-cervical sites following treatment for intraepithelial neoplasia of the cervical (CIN 2–3), arguing for the implementation of individualized follow-up and screening initiatives (Preti M, Rosso S, Micheletti L, Libero C, Sobrato I, Giordano L, et al., 2020).

Comment:

- page 11, paragraph 4.3: the cervical cancer screening visit should be an important prevention moment: also patients should be educated about vulvar self-examination for reporting minor and major vulvar changes (please cite: Preti M, Selk A, Stockdale C, Bevilacqua F, Vieira-Baptista P, Borella F, et al. Knowledge of Vulvar Anatomy and Self-examination in a Sample of Italian Women. J Low Genit Tract Dis. 2021;25(2):166–71. )

Response:

Thank you for the suggestion. We have indicated in the discussion under 4.3 thus:

Consequently, educating women would lead to increased uptake of the screening test; therefore, patients should be educated on vulvar self-examination for reporting minor and major vulvar changes, and cervical cancer screening visits should be viewed as a crucial opportunity for cancer prevention (Preti M, Selk A, Stockdale C, Bevilacqua F, Vieira-Baptista P, Borella F, et al. 2021).

Comment:

- page 11, paragraph 4.3: educational intervention in order to diminish the impact of HPV related cancers (recent multi society statements reinforced this concept: should be oriented

Preti M, Joura E, Vieira-Baptista P, et al The European Society of Gynaecological Oncology (ESGO), the International Society for the Study of Vulvovaginal Disease (ISSVD), the European College for the Study of Vulval Disease (ECSVD) and the European Federation for Colposcopy (EFC) consensus statements on pre-invasive vulvar lesions International Journal of Gynecologic Cancer  Published Online First: 21 June 2022. doi: 10.1136/ijgc-2021-003262;  Kesic V, Carcopino X, Preti M, et al The European Society of Gynaecological Oncology (ESGO), the International Society for the Study of Vulvovaginal Disease (ISSVD), the European College for the Study of Vulval Disease (ECSVD), and the European Federation for Colposcopy (EFC) consensus statement on the management of vaginal intraepithelial neoplasia International Journal of Gynecologic Cancer 2023;33:446-461.)    

 I thank the authors for this precious work.

Response:

Thank you for the suggestion. We have indicated in the discussion under 4.3 thus:

In order to reduce the impact of HPV-related malignancies, educational intervention should be centered on vaccination and screening efforts; recent multi-society statements have reinforced this concept. (Preti M, Joura E, Vieira-Baptista P, et al 2022;  Kesic V, Carcopino X, Preti M, et al 2023).

Reviewer 2 Report

Dear Authosr,

I appreciate your work and the idea of this study is really interesting. I have read your manuscript and I have the following remarks.

Tha abstract should include more concrete data about the results of the study.

In the introduction I would mention also the aim of the study.

I consider that there are two biases in methodology- the reduce number of participants and the  age interval is too wide.

From my point the review section is too narrative. I would suggest to collect data, code them and try to do a statistic report, otherwise maybe the title of the article should be ....case series

The topic of the article is very known and I think that the research can be extended in an extensive number of subjects.

The bibliography can be extended as well and more recently

Author Response

Comments and Suggestions for Authors

Dear Authors,

I appreciate your work and the idea of this study is really interesting. I have read your manuscript and I have the following remarks.

Comment:

That abstract should include more concrete data about the results of the study.

Response:

We have revised the abstract concisely.

Comment:

In the introduction I would mention also the aim of the study.

Response:

We have revised the aim of the study thus: “Therefore, the present study aims to explore the knowledge and attitudes of cervical cancer and papanicolaou (Pap) smear screenings among women in Caleb Motshabi district, Bloemfontein, South Africa”.

Comment:

I consider that there are two biases in methodology- the reduce number of participants and the age interval is too wide.

Response:

We appreciate your comment; however, interviews were conducted until saturation point (when there were no new emerging facts or information). In addition, as stated in the manuscript, we included the age bracket of 18-60 because in the context of the South African women Cervical Cancer Policy, this particular age group is legally eligible for free screening. In addition, the above age group presents with the highest incidence of cervical cancer and as well as highest disease incidence in South Africa.

Comment:

From my point the review section is too narrative. I would suggest to collect data, code them and try to do a statistic report, otherwise maybe the title of the article should be ....case series

Response:

This was a qualitative study where in-depth interviews were conducted. In the light of your comment concerning the title, we have reframed it thus: Perception of women's knowledge of and attitudes toward cervical cancer and Papanicolaou smear screenings: a qualitative study in South Africa

Comment:

The topic of the article is very known and I think that the research can be extended in an extensive number of subjects.

Response:

We have made comparison with other similar studies elsewhere.

Comment:

The bibliography can be extended as well and more recently.

Response:

We have revised the paper with some recent citations including the other references suggested by one of the reviewer. Examples of these citations include the following:

New references

Herrero R, Murillo R. Cervical cancer. In: Thun M, Linet MS, Cerhan JR, Haiman CA, Schottenfeld D, eds. Cancer Epidemiology and Prevention. 4th ed. Oxford University Press; 2018:925-946.

Zuma, K., Simbayi, L., Zungu, N., Moyo, S., Marinda, E., Jooste, S., . . . Ramlagan, S. (2022). The HIV epidemic in South Africa: Key findings from 2017 National Population-Based Survey. International Journal of Environmental Research and Public Health,19(13), 8125. doi:10.3390/ijerph19138125

Stelzle, D., Tanaka, L. F., Lee, K. K., Ibrahim Khalil, A., Baussano, I., Shah, A. S., . . . Dalal, S. (2021). Estimates of the global burden of cervical cancer associated with HIV. The Lancet Global Health,9(2). doi:10.1016/s2214-109x(20)30459-9

World Health Organization. (2020). Who releases new estimates of the global burden of cervical cancer associated with HIV. Retrieved December 31, 2022, from https://www.who.int/news/item/16-11-2020-who-releases-new-estimates-of-the-global-burden-of-cervical-cancer-associated-with-hiv

Canfell K, Shi JF, Lew JB, et al. Prevention of cervical cancer in rural China: evaluation of HPV vaccination and primary HPV screening strategies.Vaccine. 2011;29:2487-2494.

Campos NG, Sharma M, Clark A, et al. The health and economic impact of scaling cervical cancer prevention in 50 low and lower-middle-income countries. Int J Gynaecol Obstet. 2017;138(suppl 1):47 -56.

Bosch FX, Robles C, Diaz M, et al. HPVFASTER: broadening the scope for prevention of HPV-related cancer. Nat Rev Clin Oncol. 2016;13:119-132.

Preti, M.; Rosso, S.; Micheletti, L.; Libero, C.; Sobrato, I.; Giordano, L.; et al. Risk of HPV-related extra-cervical cancers in women treated for cervical intraepithelial neoplasia. BMC Cancer, 2020, 20.

Preti, M.; Selk, A.; Stockdale, C.; Bevilacqua, F.; Vieira-Baptista, P.; Borella, F.; et al. Knowledge of Vulvar Anatomy and Self-examination in a Sample of Italian Women. J Low Genit Tract Dis. 2021,25,166–71. 

Preti, M.; Joura, E.; Vieira-Baptista, P.; Van Beurden, M.; Bevilacqua, F.; Bleeker, M.C.G.; Bornstein, J.; et al. The European Society of Gynaecological Oncology (ESGO), the International Society for the Study of Vulvovaginal Disease (ISSVD), the European College for the Study of Vulval Disease (ECSVD) and the European Federation for Colposcopy (EFC) Consensus Statements on Pre-invasive Vulvar Lesions. Journal of lower genital tract disease, 2022, 26, 229–244.

Kesic, V.; Carcopino, X.; Preti, M.; Vieira-Baptista, P.; Bevilacqua, F.; Bornstein, J.; Chargari, C.; et al. The European Society of Gynaecological Oncology (ESGO), the International Society for the Study of Vulvovaginal Disease (ISSVD), the European College for the Study of Vulval Disease (ECSVD), and the European Federation for Colposcopy (EFC) consensus statement on the management of vaginal intraepithelial neoplasia. Inte J gyne cancer: official J the Inte Gyne Cancer Society, 2013, 33, 446–461.

Reviewer 3 Report

This is a qualitative study applying FGD approach to evaluate knowledge and attitude in prevention of cervical cancer. The method applies is appropriate for the study design and the findings are useful for the local context.

Introduction:

This section provides a basis for the study though the need to include issues related to HIV may not be relevant. The aim for the study is justifed.

Method.

The qualitative study method applies only FGD. Additonal tools like interview techniques and /or survey could have led to more robust results to support the comments made in the discussion. 

The recruitment of subjects from a religious centre may have some selection bias . The background of the subjects need to be elaborated to reflect on the population at risk. The specific age and level of education are influencing factors.

Both these two issues should be stated under 'limitations of the study' .

Results:

A thematic approach has been taken in addressing the areas of concer in regards to knowledge and attitude to cervical cancer prevention. It appears that only 6/19 subjects had undergone pap smears,. This is a limiation of the study as personal experiences relating to the procedure and attitude of nurses may only apply to the small group.  While the concerns expressed by the authors are applicable to a prevention program of this nature, it is vital to emphasise some of the comments ( related to the procedure of pap smear) is only drawn for this select group.  

The study shows there are concerns in the success of pap smear and its uptake . However, the authors needs to be careful in writing about a very comprehensive intervention strategy based on this study alone. 

Discussion.

The has author written about several possible strategies that are employed  in many LIC countries. A comment about their applicability and uptake in the local population would be useful for policy makers. Adding a statement that further studies e.g. cross-sectional quantitative studies or population surveys on some of the interventions suggested ,  would be useful to move forward. 

Limitations of the study: Kindly add these based on the comments above.

Author Response

Comments and Suggestions for Authors

This is a qualitative study applying FGD approach to evaluate knowledge and attitude in prevention of cervical cancer. The method applies is appropriate for the study design and the findings are useful for the local context.

Introduction:

Comment:

This section provides a basis for the study though the need to include issues related to HIV may not be relevant. The aim for the study is justified.

Response:

The issue of HIV was reflected in providing the context of the study because research has linked it as one of the cofactor for the development of cervical cancer (Herrero & Murillo, 2018). In fact, the global prevalence of new HIV-attributable cervical cancer among women living with HIV (WLHIV) is 6% (Stelzle et al. 2021, WHO, 2020).  In addition, in the proportion of women living with HIV in South Africa is seemingly high (51%) (Zuma et al., 2022), a statistic implying that it is a high-risk environment for the acquisition of cervical cancer. Nonetheless, we have revised this aspect in the introduction as highlighted above.

Method.

Comment:

The qualitative study method applies only FGD. Additional tools like interview techniques and /or survey could have led to more robust results to support the comments made in the discussion. 

Response:

Unfortunately these additional aspects were not included in the study and thus highlighted as limitations of the study.

Comment:

The recruitment of subjects from a religious centre may have some selection bias. The background of the subjects need to be elaborated to reflect on the population at risk. The specific age and level of education are influencing factors. Both these two issues should be stated under 'limitations of the study'.

Response:

The subjects were not recruitment from a religious centre, rather the church was used as the venue for the focus group interviews to ensure there was ample space and minimal interference.

Results:

A thematic approach has been taken in addressing the areas of concern in regards to knowledge and attitude to cervical cancer prevention. It appears that only 6/19 subjects had undergone pap smears. This is a limitation of the study as personal experiences relating to the procedure and attitude of nurses may only apply to the small group.  While the concerns expressed by the authors are applicable to a prevention program of this nature, it is vital to emphasise some of the comments (related to the procedure of pap smear) is only drawn for this select group.  

Response:

Thank you for this observation. Our (authors) comment regarding the healthcare professionals is not necessary and the sentence has been deleted; only the views of the women were sought, and not the nurses.

Comment:

The study shows there are concerns in the success of pap smear and its uptake. However, the authors needs to be careful in writing about a very comprehensive intervention strategy based on this study alone. 

Response:

This is noted. However, the proposed interventions were also based on the suggestions by the participants regarding their challenges about the uptake of cervical and pap smear screening. Therefore, the interventions were context-specific as highlighted in the paper under section 4.5.

Discussion.

Comments:

The has author written about several possible strategies that are employed in many LIC countries. A comment about their applicability and uptake in the local population would be useful for policy makers. Adding a statement that further studies e.g. cross-sectional quantitative studies or population surveys on some of the interventions suggested, would be useful to move forward. Limitations of the study: Kindly add these based on the comments above.

Response:

We have added a statement based on the comment above and the limitation of the study.

Limitations of the study

The limitations of this study are worth noting. Caleb Motshabi was the only location in Bloemfontein where the study was administered; consequently, all participants were from this location. Due to the small sample size and the fact that only one race participated in this study, it is not possible to assume that all races in this context share the same perspectives. Furthermore, these results cannot be extrapolated to the entire Free State Province nor to the entire country.  In addition, considering that the present research was a qualitative investigation that aimed to provide insights, additional quantitative cross-sectional studies or population surveys as well as context-specific strategies about cervical cancer screening are required. Despite the dearth of available data, this study provides some insight into the perspectives of cervical cancer screening in a region with scarce resources. 

Reviewer 4 Report

The authors present a qualitative assessment of cervical cancer screening knowledge and attitudes among South African women, in order to drill down on obstacles and gaps and identify possible interventions to improve screening uptake. 

The topic is important and the methodology is appropriate for this type of study. I wonder whether the findings are transferrable to other South African communities including other townships, urban and rural communities. If the authors give some information in terms of subjects' demographics, social and educational background and compare the characteristics of the community sampled here to other South African communities, it may round out their case.

There are several errors in the abstract and introduction that require correction:

line 12 - cervical cancer is not the most common cancer globally

lines 39-44 - authors fail to list smoking as a risk factor for cervical cancer

line 47 - HPV vaccine availability is unrelated to the likelihood of recovery from early cervical cancer

line 57 - 4.7 million women living with HIV in South Africa are not 62%

line 69 - these factors are mediating rather than mitigating factors

Finally, in order to clarify and streamline the discussion, particularly in the section on recommendations (beginning line 297), I would suggest grouping the gaps identified and the ensuing suggestions into topics. For example, 1. Education and awareness (including school and community education efforts, awareness campaigns, etc.) 2. Health services organization (including clinic organization, dedicated women's health days, mobile clinical and transportation) 3. Health services provision (including provider attitudes, patient education, opportunistic screening etc.)

The language is generally good, but some English language editing would be useful

Author Response

Comments and Suggestions for Authors

The authors present a qualitative assessment of cervical cancer screening knowledge and attitudes among South African women, in order to drill down on obstacles and gaps and identify possible interventions to improve screening uptake. 

Comments:

The topic is important and the methodology is appropriate for this type of study. I wonder whether the findings are transferrable to other South African communities including other townships, urban and rural communities. If the authors give some information in terms of subjects' demographics, social and educational background and compare the characteristics of the community sampled here to other South African communities, it may round out their case.

Response:

We appreciate your valuable suggestion. The demographic profile of the participants has been included in the result section.

3.1 Demographic profile of the participants

The age of the participants ranged from 18-60 years. The age group was selected because cervical cancer is often diagnosed in middle aged women but rarely diagnosed in women under the age of 20 years (WHO, 2014). Of the 19 participants, four participants possessed a high school qualification and none had a tertiary education, most single, and almost half of the participants were employed on short term contract

Table 1. Demographic profile of the participants (n=19)

Variables

Number of participants

Age (years)

20-30

3

31-45

11

45-60

5

Level of education

Grade 7-9

8

Grade 10-11

7

Grade 12

4

Marital status

Married

3

Single

15

Widowed

1

Employment status

Employed

7

Unemployed

12

Comment:

There are several errors in the abstract and introduction that require correction:

Response:

One of the reviewers has also alluded to this; therefore, we have revised the abstract concisely.

Comment:

line 12 - cervical cancer is not the most common cancer globally

Response:

Thank you this observation; this has been corrected to read thus:

According to worldwide statistics, cervical cancer is the fourth most commonly diagnosed cancer and the fourth primary cause of cancer-related mortality in women, with an approximated 604, 000 cases newly identified and 342,000 deaths in 2000 (Sung et al., 2021). Based on global data, cervical cancer is the cancer that is diagnosed the most often and the fourth main cause of cancer-related death in women, with an estimated 604,000 cases being diagnosed and 342,000 deaths in 2020 (Sung et al., 2020). Also, in contrast to developed nations, females in less developed countries have substantially higher cervical cancer mortality rates (12,4 per 100,000 versus 5,2 per 100,000) (1Sung et al., 2021). On a regional basis cervical cancer is a major cause of cancer-related mortality in 36 nations, and nearly all of these countries are located in sub-Saharan Africa, including the Republic of South Africa [1].

Comment:

lines 39-44 - authors fail to list smoking as a risk factor for cervical cancer

Response:

We have added other significant precursors include certain sexually transmitted infections (HIV and Chlamydia trachomatis), smoking, and long-term oral contraceptive usage (Herrero & Murillo, 2018).

Comment:

line 47 - HPV vaccine availability is unrelated to the likelihood of recovery from early cervical cancer

esponse:

We have revised the sentence to read thus:
…owing to the extremely effective primary (HPV vaccine) and secondary (screening) preventative measures, cervical cancer is thought to be essentially preventable (Sung et al., 2021)

Comment:

line 57 - 4.7 million women living with HIV in South Africa are not 62%

Response:

This percentage has been corrected to 51% and reference cited accordingly. “Furthermore, 51% of South African women are living with HIV/AIDS (Zuma et al., 2022),…

Comment:

line 69 - these factors are mediating rather than mitigating factors

Response:

This has been corrected to mediating factors.

Comments:

Finally, in order to clarify and streamline the discussion, particularly in the section on recommendations (beginning line 297), I would suggest grouping the gaps identified and the ensuing suggestions into topics. For example, 1. Education and awareness (including school and community education efforts, awareness campaigns, etc.) 2. Health services organization (including clinic organization, dedicated women's health days, mobile clinical and transportation) 3. Health services provision (including provider attitudes, patient education, opportunistic screening etc.)

Response:

Thank you for the above suggestion. We have revised the section accordingly thus:

Participants reported negative experiences when visiting clinics for cancer or other medical appointments; these negative experiences could be viewed as potential barriers to their adoption of cervical cancer screening and vaccination.  The combined vaccination and screening strategy has been shown to be cost-effective in a number of low-middle income countries (Cantel et al., 2011; Compos et al., 2017; Bosh et al., 2016), suggesting that it may be possible to prevent cervical cancer in contexts with limited resources. As previously stated, the majority of the participants in this present study cited challenges relating to education and awareness (lack of school and community education efforts, awareness campaigns), health services organization (inadequacies related to poor clinic organization, lack of dedication to women's health days, lack of mobile clinics and transportation), as well as poor quality of health services provision (negative and poor attitudes of healthcare providers). However, the majority of problems are due to human error and therefore can be addressed by adopting appropriate measures in the clinical setting. Therefore, a collaborative and multi-sector strategy is needed to mitigate these challenges.  Government, schools, and non-governmental organizations concerned with women's health should create health education advocacy and awareness in schools and communities, as well as organize health talks and road shows on the significance of cervical cancer screening. In this context, the Department of Health ought to strengthen the existing community-wide awareness campaigns for additional illnesses and integrate cervical cancer awareness into its awareness program.  Additionally, the educational programs, whether administered in clinics or as outreach programs, will raise community awareness about cervical cancer and emphasize the risk factors associated with the disease. Furthermore, the availability of sufficient resources, as well as trained and committed personnel, would aid in promoting cancer screening. Finally, considering that this was a qualitative study that sought to provide insights, further studies that include cross-sectional quantitative studies or population surveys on some of the interventions proposed here to address the context-specific challenges of cervical cancer screening in this low-resource geographical setting would provide more comprehensive and accurate perspectives on the effectiveness or implementation of these interventions.  .  

Comments on the Quality of English Language

The language is generally good, but some English language editing would be useful
